# Assessing Polysaccharides/Aloe Vera–Based Hydrogels for Tumor Spheroid Formation

**DOI:** 10.3390/gels9010051

**Published:** 2023-01-07

**Authors:** Petruța Preda, Ana-Maria Enciu, Cristiana Tanase, Maria Dudau, Lucian Albulescu, Monica-Elisabeta Maxim, Raluca Nicoleta Darie-Niță, Oana Brincoveanu, Marioara Avram

**Affiliations:** 1National Institute for Research and Development in Microtechnologies—IMT Bucharest, 126A Erou Iancu Nicolae, 077190 Bucharest, Romania; 2Biochemistry-Proteomics Department, Victor Babes National Institute of Pathology, 99-101 Splaiul Ind Pendentei, 050096 Bucharest, Romania; 3Cell Biology and Histology Department, Carol Davila University of Medicine and Pharmacy, 8 Eroii Sanitari, 050474 Bucharest, Romania; 4Faculty of Medicine, Titu Maiorescu University, Gheorghe Petrascu St., 031593 Bucharest, Romania; 5“Ilie Murgulescu” Institute of Physical Chemistry, Romanian Academy, Spl. Independentei 202, 060021 Bucharest, Romania; 6Physical Chemistry of Polymers Department, Petru Poni Institute of Macromolecular Chemistry, 41A Grigore Ghica Voda Alley, 700487 Iasi, Romania

**Keywords:** hydrogel, natural polymer, polysaccharides, tumor spheroids, aloe vera gel

## Abstract

In vitro tumor spheroids have proven to be useful 3D tumor culture models for drug testing, and determining the molecular mechanism of tumor progression and cellular interactions. Therefore, there is a continuous search for their industrial scalability and routine preparation. Considering that hydrogels are promising systems that can favor the formation of tumor spheroids, our study aimed to investigate and develop less expensive and easy-to-use amorphous and crosslinked hydrogels, based on natural compounds such as sodium alginate (NaAlg), aloe vera (AV) gel powder, and chitosan (CS) for tumor spheroid formation. The ability of the developed hydrogels to be a potential spheroid-forming system was evaluated using MDA-MB-231 and U87MG cancer cells. Spheroid abilities were influenced by pH, viscosity, and crosslinking of the hydrogel. Addition of either AV or chitosan to sodium alginate increased the viscosity at pH 5, resulting in amorphous hydrogels with a strong gel texture, as shown by rheologic analysis. Only the chitosan-based gel allowed formation of spheroids at pH 5. Among the variants of AV-based amorphous hydrogels tested, only hydrogels at pH 12 and with low viscosity promoted the formation of spheroids. The crosslinked NaAlg/AV, NaAlg/AV/glucose, and NaAlg/CS hydrogel variants favored more efficient spheroid formation. Additional studies would be needed to use AV in other physical forms and other formulations of hydrogels, as the current study is an initiation, in evaluating the potential use of AV gel in tumor spheroid formation systems.

## 1. Introduction

Biomedical scientists are currently using three-dimensional (3D) cell cultures in the study of drug resistance and the mechanism of cancer because these systems show characteristics able to mimic in vivo conditions better than two-dimensional (2D) cell cultures [1,2,3,4]. Development of 3D cell cultures can be accomplished out in liquid-based systems, bioreactor systems, and tissue extracellular matrix (ECM)-like systems (such as porous scaffolds, fibrous scaffolds, and gel-based scaffolds) based on polymeric biomaterials. The polymeric biomaterials used must facilitate cell adhesion, proliferation, and migration, and these materials can be designed from natural polymers and/or synthetic polymers (alginate, hyaluronic acid, chitosan, gelatin, collagen/polycaprolactone, polylactic acid, polyethylene glycol, polydimethylsiloxane, etc.) [4].

Due to mimicking native tumor tissue, multicellular tumor spheroids (MCTS) are the current approach used in vitro as 3D cancer-cells-culture models for therapeutic strategies (e.g., drug therapy testing, studies of tumor progression under different conditions such as early stage and avascular tumors, and also to evaluate the release of soluble mediators that provide clues for immunosurveillance) [5] and genetic screening and editing of tumor cells for personalized medicine [6,7].

MCTS can be easily grown in a liquid suspension culture, but it is difficult to control their size and uniformity with this method. Alternatively, more uniform-sized spheroids are generally developed by culturing cells in a confined space, such as suspended liquid droplets and porous/fibrous/gel-based hydrogel scaffolds [8,9,10,11]. Hydrogel scaffolds (artificial hydrogel-based ECMs) used in the development of 3D cell culture are limited due to the supply of oxygen, nutrients to internal cells, and transport of metabolic wastes, but the development of scaffolds with adequate porosity and promoting agents (such as growth factors) could be promising [5,12,13,14]. In addition, due to these diffusion constraints, the size of spheroids is limited to 200–500 µm [4,15,16].

Ideal scaffold hydrogels should be developed from degradable, biocompatible, mechanically tunable (e.g., in shape, stiffness, or porosity), and cost-effective polymeric materials [17,18,19]. Scaffold hydrogels can be engineered to mimic a series of features of native ECMs. Protein-based hydrogels such as collagen, Matrigel, and fibrin are frequently used as 3D cancer-cell-culture development due to their biophysical properties and specific cell adhesion [20].

Matrigel^®^ is considered the “gold standard” in the development of 3D cell cultures due to its properties of cell attachment, morphogenesis, proliferation, and organization. It is composed of collagen IV, laminin, fibronectin, entactin, and growth factors. Through polymerization, Matrigel forms a dense gel with small pores. Many studies reported the successful growth and evaluation of 3D cell cultures using Matrigel^®^; its drawback being the high price for regular usage. As a result, there is a need for a more affordable alternative for 3D-cell-culture hydrogel scaffolds [19]. Recently, numerous hydrogels have been studied, to find more advantageous formulations for tumor-spheroid formation, which would be less expensive and overcome diffusion constraints. Hence, the potential of oxygenating microgels formed using perfluorocarbon (PFC) modified chitosan was studied by Patil et al., which facilitated the growth of larger human cell-based spheroids with higher oxygen pressures into spheroids [21]. Furthermore, Vignesh et al. demonstrated that novel thermoresponsive copolymers based on poly(N-isopropylacrylamide) (p(NIPA)) can be used as additives for the rapid formation of spheroids, via the hanging drop method [16]. Hydrogels based on alginate/gelatin/Matrigel were developed as bioprinting bioinks for spheroid formation and subsequently reprinted to generate multigenerational models. The study proved that immortalized triple-negative breast cancer cells (MDA-MB-231) and patient-derived gastric adenocarcinoma cells can be reprinted for at least three cycles of the 21-day culture [22].

Alginate was also mixed with Matrigel to investigate the malignant progression of normal mammary epithelium mediated by the changing hydrogel stiffness [23]. Recently, human ovarian-cancer-cell-spheroid formation was reported in alginate–collagen–agarose hydrogel [24].

Unlike Matrigel or collagen, polysaccharide-based hydrogels provide controllable, stable chemical and physical conditions for the growth of tumoral spheroids, as their degradation is independent of proteolytic enzymes secreted by cells [25]. Based on these data, the aim of the present study was to investigate whether AV would be a useful addition to alginate and chitosan-based hydrogels, to generate less expensive, ease-to use, and reliable scaffolds for tumor spheroid formation.

## 2. Results and Discussion

In the present research, amorphous and solid (crosslinked) hydrogels were developed based on natural compounds such as alginate, aloe vera (AV) gel powder, and chitosan as potential systems for the development of tumor spheroids. The current approach started from our previous studies made for such gels, but evaluated for their applicability in the field of wound healing [26]. Therefore, the aim of this study was to investigate another potential direction of applicability of amorphous hydrogels and their solid version containing alginate and aloe vera, respectively. It should be mentioned that the formulation of the hydrogels in the current research was adapted with small changes based on the previous study. The use of the alginate, AV, and chitosan in the developed biocomposites is due to their valuable properties, especially their biocompatibility, solubility in non-toxic solvents, hydrophilicity, promotion of cell adhesion, low toxicity, minimal immune response, and accessibility [27,28]. Another consideration in the development of hydrogels based on natural polymers was based on the fact that this type of hydrogel mimics native ECM architecture [20].

Alginate is a natural polysaccharide polymer, extracted from brown algae, and previously used in drug delivery, wound dressing, tissue engineering, and 3D-cell-culture setups. Starting from the aqueous solution of alginate, hydrogels can be obtained by ionic and covalent crosslinking [29].

The addition of AV in our hydrogel variants for the formation of tumor spheroids was based on its proliferative properties, due to phytochemical compounds such as mucopolysaccharides, vitamins, minerals, enzymes, proteins, amino acids, sterols, and salicylic acid. These bioactive compounds found in the AV gel (inner parenchyma of the leaf) promote cell attachment, migration, proliferation, and cell development [30].

To our knowledge, there is only one study that follows this direction, exploiting the potential of acemannan-based hydrogel as system for the development of tumor spheroids [31]. Acemannan is the polysaccharide present in AV gel that has the same biological activity, including the antitumor effect, as that mediated by immunomodulation in vivo [32].

Chitosan is a biocompatible, biodegradable, non-immunogenic, hydrophilic natural polymer soluble in diluted acids [33,34,35]. The linear polycationic polysaccharide structure composed of randomly distributed D-glucosamine and N-acetyl-glucosamine monomers of chitosan is similar to glycosaminoglycans (the main components of the tumor extracellular matrix) [36,37]. Kievit et al. developed chitosan–alginate scaffolds by lyophilizing and crosslinking a physical mixture for cell attachment and proliferation, ideal for modeling the tumor microenvironment. The scaffold allowed the proliferation of U-87 MG and U-118 MG human glioma tumor cells in vitro for 10 days [38]. Furthermore, other studies based on the chitosan scaffold have been reported as a more reliable in vitro cell culture platform than traditional 2D culture, e.g., chitosan-based scaffolds with different natural compounds to mimic ECM, such as collagen [39], hyaluronic acid [40], or pectin [41].

The developed biocomposites were achieved with different pH and based of alginate to which aloe vera and/or chitosan, and/or glucose, were added. Glucose appears at the top of the metabolic hierarchy of cancer cells [42] and supports the growth and survival of tumor cells [43]. The obtained hydrogels at low acid pH were started from the idea that the tumor-cell microenvironment is acidic [44]. Therefore, gels with pH 5 were synthesized, to mimic this acidity. On the other hand, basic pH (pH 12) hydrogels were synthesized in order to investigate whether neutralization of this acidity would be beneficial for cells, or whether it would influence spheroid formation. Furthermore, the pH influences the physical texture of the gel. It was observed that at pH 5 and 2–8 °C, the polymer biocomposites acquired a gel texture following the addition of AV solution, which was not observed in the pH 12 hydrogels. This remark was visually observed and was also confirmed by rheological and FT-IR analysis.

### 2.1. Structural and Rheological Investigation of Hydrogels

The biologically significant hydrogels were evaluated by FT-IR analysis, to highlight the changes in the structure of the composites due to the pH and the crosslinking with a 1% calcium chloride solution. All spectra (Figure 1) showed a broad absorption band due to the stretching vibrations of hydroxyl groups (-OH). This band was centered at 3253 cm^−1^ for sodium alginate and is shifted at lower wave numbers (3242 cm^−1^) for the V2 and V4 hydrogels at pH 12. The shift decreased more (3231 cm^−1^) at pH 5 for V1.

A low intensity peak around 2930 cm^−1^ attributed to alkyl C-H stretching vibration was found in all spectra (Figure 1). The absorption bands at 1594 cm^−1^ and 1408 cm^−1^ were assigned to asymmetric and symmetric stretching vibration of carboxylate anions (COO-) in the sodium alginate spectrum [45,46]. These absorption bands decreased in wave numbers for all spectra, and can be attributed to interaction between sodium alginate and the AV and chitosan. The absorption band at 1081 cm^−1^ can be attributed to the C-O stretching vibration from sodium alginate molecule and AV [47]. This peak was shifted at 1077 cm^−1^ because of endocyclic C-O bonds from glucose introduced into the biocomposite V4. The peak around 1026 cm^−1^ was due to the C-O-C stretching vibration in glycosidic bonds of sodium alginate molecule and in glucan units from AV [48,49]. The absorption peaks at 949 cm^−1^, 887 cm^−1^, and 817 cm^−1^ were specific to the guluronic and mannuronic acid sequences in the sodium alginate molecules [49]. In the biocomposites with AV, the peak at 887 cm^−1^ was shifted to 875 cm^−1^, 871 cm^−1^, and 870 cm^−1^ due to C-H out-of-plane deformation in mannose present in the AV gel powder [47].

Chitosan molecule characteristic bands can be observed for V5 (Figure 1) at 1636 cm^−1^ assigned to amide I (stretching vibration of C = O bond with N-H deformation), 1022 cm^−1^ for C-O stretching vibration, 1070 cm^−1^ corresponding to stretching vibration of glycosidic linkage C-O-C, and at 893 cm^−1^ for C-H out-of-plane deformation [50].

Therefore, the increase in the intensity of the band corresponding to the hydroxyl group, as well as its shift when adding AV to alginate, suggest possible chemical interactions. In the case of the hydrogel with pH 5 (V1), the displacement decreased a lot, probably due to the protonation of some compounds from AV and the occurrence of strong electrostatic reactions between them. There might be a similar explanation in the case of the V5 hydrogel variant with pH 5; it is possible to protonate the amino group from the chitosan structure and implicitly achieve electrostatic interaction with alginate, thus being able to create a hydrogel with a strong gel texture (proved by rheological).

The absorption bands corresponding to the stretching vibration of O-H bond in all crosslinked hydrogels and sodium alginate (NaAlg) with calcium ions (Figure 2) appeared narrower than those of the uncrosslinked gels [51] and are shifted at higher wave numbers. In addition, a shift of the peaks corresponding to the COO- carboxylate groups from crosslinked gels spectra comparing to those for uncrosslinked gels indicated an ionic binding between sodium alginate guluronate blocks and Ca^2+^ ions [52].

Rheological CSR tests of the gels that proved to be potential systems for the development of spheroids were performed and the observations related to viscosity showed different behaviors of gels with different compositions at different pH (Figure 3a).

A Newtonian plateau was found for NaAlg for almost all shear rate domains, with a very slight shear-thinning behavior at the highest tested shear rates. A similar behavior has been reported for low molecular weight alginates intended to be used in 3D printing [53]. Except for the neat NaAlg (2%), all evaluated hydrogel formulations demonstrated shear-rate dependence and pseudoplastic (shear-thinning) behavior, with viscosity decreasing as the shear rate increased.

The pH has a strong effect on the flow behavior of the studied hydrogel samples. The hydrogels prepared at acidic pH (pH 5) demonstrated higher viscosity at low shear rates compared to the sample at basic pH (pH 12). The probable reason could be the polarization of bonds between hydroxide and carboxylate bonds at lower pH, enhancing the polar interactions between the components that leads to the formation of a stiffer hydrogel [54]. This information is consistent with the information provided by the FT-IR analysis by the shift of the spectral bands attributed to hydroxyl groups.

At pH 5, the hydrogels with AV or chitosan (V5) showed an increased viscosity compared with NaAlg, especially as the shear rate was very low or approached zero, suggesting that the gel does not flow at rest [55]. Steady-state flow curves of V1 and V3 gels at pH 5 showed a slight rise in viscosity at low shear rate, possibly because the hydrogels tend to resist the flow due to the strong interactions between hydrogel constituents. Similar behavior has been reported by Irfan et al. when evaluating the flow behavior of xanthan gum, acrylic acid, and N-isopropyl acrylamide [56].

With further increase in shear rate, the hydrogels chains at pH 5 are forced to flow in the direction of the applied shear rate, indicating shear-thinning of a non-Newtonian nature (Appendix A). The hydrogel V5, which contains chitosan, is the stiffer material among the hydrogels obtained at pH 5. An increase of pH from 5 to 12 led to the loss of the Newtonian plateau and a sharp decrease of viscosity for V2 and V4 of about two orders of magnitude at low shear rates, but with a smaller dependence of flow function of deformation. The decrease in viscosity as a function of shear rate was steeper for stiffer hydrogels at pH5, compared with the slower decrease for the hydrogels V2 and V4 at pH 12.

The amplitude sweeps tests showed information concerning the effect of hydrogel components and pH on the samples’ structures. The dynamic strain sweeps were conducted at variable amplitudes (strain) and constant frequency to identify the linear viscoelastic (LVE) region for each hydrogel in the region where G’ results were independent of the applied deformation. The amplitude dependence of the dynamic moduli resulted for alginate hydrogels presented in Figure 3b denoted that most of the formulations showed a linear behavior of G’ up to about 1% strain, with this value for constant strain being used further for frequency sweep tests (within LVE range). Outside the LVE, G’ decreased indicating structure break-down due to plastic deformations.

Neat NaAlg showed liquid-like behavior, with G” > G’ over the entire evaluated deformation range. Similar behavior has been reported by the authors for alginate-based hydrogels containing type I collagen isolated in-house from silver carp tails [26].

Strain sweeps showed gel-like behavior for hydrogels obtained at acidic pH, since, for all these samples, the elastic modulus G’ was higher than the viscous modulus G”. Both hydrogels containing AV, with or without glucose obtained at pH 5 maintained their stable gel structure independent of the tested deformation strain range.

The hydrogels realized with a basic pH (pH 12) changed their structure from gel to liquid-like between 1 and 5% applied strain, when dynamic moduli crossover could be observed. In addition, the pH had a strong influence on the viscoelastic properties of the evaluated hydrogels—the lower (more acidic) the pH, the higher the strength of the resulting developed hydrogels (Appendix A).

Both hydrogels containing AV solution and chitosan obtained at pH 5 maintained their stable gel structure independent of the tested deformation strain range. Acidic pH led to stronger gels.

### 2.2. Evaluation of Spheroid Formation

Both amorphous and crosslinked hydrogels (solid hydrogels) were tested for spheroid formation, using two aggressive cell lines (a breast carcinoma cell line—MDA-MB-231 and a glioblastoma cell line—U87). These lines were selected in this study because our previous work with MDA-MB-231 in Matrigel showed that they tend to enzymatically degrade it over time, generating a mixed 2D and 3D cell culture (Appendix A). Similarly, U87 cells seeded in a gradient hydrogel plate degraded the low-density hydrogel, forming 2D cultures, while growing in spheroids in the high-density hydrogel (Appendix A). After testing different commercial spheroid solutions, three methods were used as reference in present study, two based on hydrogels (with and without an adhesion peptide, to favor cell attachment) and an ultralow attachment plate. Among all developed hydrogel variants only V2, V4, and V5 developed spheroids for both cell lines (Figure 4), while V1, although allowed proliferation of cells, did not generate spheroids (Appendix A). The fact that the amorphous and solid-gel variants based on NaAlg and AV gel-powder solution support cell development, even without spheroid formation in the case of V1 sample, suggested that aloe vera used in the biocomposite had no cytotoxic effect against the tested tumor cells.

As previously mentioned, both the amorphous and the solid hydrogels were tested and it should be noted that the solid gels were more efficient in generating spheroids than their amorphous counterparts (Figure 4). It was also observed that the AV-based variants with pH 5, did not facilitate the formation of spheroids in both crosslinked and amorphous form (Appendix A). A hypothesis of this behavior would be that the strong gel texture (proved by rheology) of the hydrogels could influence oxygen diffusion, as well as cellular mobility.

The selected crosslinked hydrogels were further tested for cellular viability using live/dead fluorescent assay, where living cells incorporated the green fluorescent dye and the dead cells the red fluorescent dye (Figure 5). As in any healthy cell culture, the majority of cells were alive in the tested gels. In addition, especially for U-87 MG, dead cells were found within the spheroids, which recapitulates the uneven nutrient and oxygen distribution within an in vivo tumor.

The V4 hydrogel was particularly efficient for U87 spheroid formation, with viable cell aggregates (with green fluorescence) ranging from 100 to 400 µm being observed (Figure 5). V5 hydrogel was also useful for generation of U87 spheroids in the range of 10 to 100 µm. The MDA-MB-231 cell line formed small cell aggregates, in line with its metastatic abilities. V2 hydrogel allowed both cell lines to form small, yet viable, spheroids. In conclusion, these three formulations showed spheroid-forming properties, although the cell type could influence the size and distribution of cells.

Previous studies reported use of alginate [57,58], or alginate–chitosan scaffolds [59] to generate viable glioblastoma spheroids. Although the number of seeding cells was much lower in the present study (10^4^, as opposed to 5 × 10^4^–10^6^), the spheroid size was larger. The use of AV could provide additional benefits for cells, as it has been reported to have bioactive components which favor cell proliferation, migration, and angiogenesis [60,61]. In addition, it has recently been demonstrated that glucomannan from AV gel facilitated the regeneration of intestinal epithelial cells [62].

Alginate has been proven as a viable solution also for MDA-MB-231 spheroids [63], but chitosan–alginate scaffolds are only beginning to be investigated [64]. So far, no data regarding alginate–AV scaffolds has been reported regarding MDA-MB-231 and U87MG spheroid formation.

Following the biological investigations and based on the results obtained, another step was to investigate the morphology of solid hydrogels with the potential to form spheroids to observe if there was any correlation between porosity and spheroid formation, using scanning electron microscopy (SEM).

V2 and V4 hydrogels with AV had no significant porosity that could be observed from SEM images (Figure 6), but showed good spheroid-forming properties independence of porosity. For the crosslinked V5 hydrogel the formation of spheroids can be associated with the porous morphology versus the non-porous morphology observed for the amorphous variant (Figure 7). In addition, the presence of calcium in crosslinked hydrogels could contribute to the spheroid formation through Ca-dependent cell junctions, such as cadherin junctions [65].

## 3. Conclusions

Among all hydrogel formulations tested, two containing AV (V2 and V4) and one variant containing chitosan (V5), in both amorphous and crosslinked forms, showed spheroid-forming abilities. However, the reticulated form performed better than the amorphous one. It should also be noted that the solid hydrogel (V4) based on alginate which contained AV and glucose (0.3%/gel formulation) potentiated the formation of spheroids better than its solid gel counterparts, especially for the U87MG cell line, developing spheroids of 100–400 µm. The next solid hydrogel that favored the formation of spheroids was alginate/chitosan hydrogel (V5); the U87MG spheroids also developed better, but with a size around of 100 µm. Finally, the V2 hydrogel allowed both cell lines to form small and viable spheroids. The V1 sample containing sodium alginate and AV allowed cell proliferation even though the gel did not potentiate spheroid formation. This indirectly suggested that AV powder in the amount used in the experimental variants does not induce a cytotoxic effect against the tested tumor cells.

Moreover, it was observed by rheological analysis that the hydrogel texture was pH-dependent. The fact that the variants containing AV at pH 12 (liquid-like) favored spheroid formation (for U87 and MDA-MB-231 cells) compared to their counterparts at pH 5, could be associated with the strong gel texture of the latter ones, which probably did not allow oxygen access and cell migration. Although it presented a gel texture, in the alginate and chitosan mixture (pH 5) crosslinking with CaCl_2_ produced a porous hydrogel, so that in this case, the porosity facilitated better formation of spheroids compared to the amorphous version.

Therefore, the incorporation of AV at different pH values was observed to induce other properties of the developed hydrogels, and the potential use of AV as a modulating factor of the ECM characteristics for spheroid-formation systems was also demonstrated. Standardization of the amount of AV gel powder in hydrogels is a promising direction for future studies, considering that the chemical composition of the AV gel may vary from batch to batch in production.

## 4. Materials and Methods

The following chemical agents were used to develop the hydrogels: sodium alginate (NaAlg) medium molecular weight (Sigma Aldrich, St Louis, MO, USA, viscosity of 2334 cps for 2% solution in water at 25 °C), NaOH (EMPLURA^®^, Darmstadt, Germany), CaCl_2_ (99, 5 %, Merck KGaA, Darmstadt, Germany), Aloe vera (AV) powder (MAYAM, Berlin, Germany), chitosan (CS) low molecular weight (Sigma Aldrich, St. Gallen, Switzerland, viscosity of 113 cps for 1% solution in 1% acetic acid), acetic acid (Chimreactiv, Bucharest, Romania), and glucose (Sigma Aldrich, St Louis, MO, USA). Culture media specific to biological tests and standard tumor cell line were also purchased.

### 4.1. Hydrogel Formulation

The gel composites were developed by stirring mixtures based on sodium alginate solution (2%), chitosan solution (1%), AV gel-powder solution (1%), and glucose solution (1%), respectively. The mixtures were obtained at room temperature (25 °C), the acid pH (pH 4.5–5) of the biocomposites (gels) was corrected with 3M NaOH solution. The obtained formulations were sterilized by exposure to UV lamp for 30 min and stored in a refrigerator (2–8 °C) until further use. The composition of developed hydrogels is presented in Table 1. It should be noted that all hydrogel variants were obtained both as amorphous and solid gels by crosslinking with CaCl_2_ solution (1%).

The preparation of the 2% NaAlg solution was carried out in deionized water by stirring for 2 h at 300 rpm and 80 °C. Similar conditions were used for obtaining 1% CS solution in 1% acid acetic. AV solution (1%) was obtained by solubilization of commercial AV powder in deionized water, followed by gentle stirring for 10 min at room temperature.

From more than 20 variants of formulations, with different compositions and pH, seven alginate-based hydrogels were selected, combined in different proportion with AV and/or chitosan, and/or glucose.

Both amorphous and crosslinked gels (with 1% CaCl_2_ solution) were obtained. All forms of hydrogels were transparent, with a slight tendency to yellow in the case of gels at pH 12. First, 2 mL of gel was distributed in 24-well sterile plates followed by addition of CaCl_2_ solution. After 30 min of CaCl_2_ solution action the crosslinked hydrogels were washed three times with deionized water. Hydrogels were sterilized by 1 h exposure to a UV-C lamp, then kept at 2–8 °C degrees until further use.

### 4.2. Structural, Morphological, and Reological Investigation of Hydrogels

The preparation of the amorphous and solid hydrogels for the morphological and structural analysis consisted in drying the samples at room temperature for 5 days. For the amorphous hydrogel variants, a film was obtained by depositing the gel on the glass slide and then drying. Samples for the crosslinked hydrogels were also obtained by drying in the wells. The morphological analysis of the obtained hydrogels was performed by using the field-emission scanning electron microscope (FE-SEM) Nova NanoSEM 630 (FEI Company, Hillsboro, OR, USA). The Fourier-transform infrared spectra (FT-IR) of obtained hydrogels were collected by using the ATR device of the Nicolet i-S10 FT-IR spectrometer (Thermo Scientific, Waltham, MA, USA). The device was equipped with a diamond crystal and the spectra were recorded in the 4000–525 cm^−1^ spectral region at 4 cm^−1^ resolution with 32 scans.

The rheological properties of the amorphous gels were evaluated by using a controlled stress rheometer Physica MCR 301 (Anton Paar, Graz, Austria) equipped with parallel-plate geometry (upper diameter of 25 mm). A Peltier system was employed to control temperature and evaporation, thus maintaining a constant hydration of the sample during testing. Steady shear and dynamic studies were realized at 37 °C. The gels were applied to the lower plate, then the upper plate was moved to a 0.5 mm gap and the excess material was removed to completely cover the upper plate. All formulations were allowed to equilibrate for at least 1 min before analysis. Different rheological measurements were performed to study the flow behavior and viscoelastic properties of the elaborated gels, i.e., one rotational, such as controlled shear rate (CSR) test to evaluate the dependence of the shear stress and viscosity on shear rate and two oscillatory tests, such as amplitude sweep test at 1 Hz, to evaluate the linear viscoelastic region (LVE) and frequency sweep tests with a previously determined fixed strain of 1% (within LVE region), in the frequency range from 0.1 to 100 rad/s.

### 4.3. Evaluation of Spheroid Formation

*Cell lines and spheroid culture*—Human breast carcinoma cells MDA-MB-231 (ATCC: HTB-26) and human glioblastoma cells U-87 MG (ATCC HTB-14) were routinely cultured in an incubator, at 5% CO_2_ and 37 °C. Every other day medium was changed with RPMI-1640 supplemented with 10% fetal bovine serum or DMEM/F12 supplemented with 10% fetal bovine serum. For spheroid formation, cells were trypsinized, centrifuged, and counted and 10,000 cells were incubated in each well containing various gel compositions.

*Spheroid protocol*–Custom-made gels were compared for spheroid formation properties with standard hydrogels, as previously described [66]. A thiol-reactive polymer (True 1, Sigma-Aldrich, St Louis, MO, USA) and a dextran-based hydrogel without RGD peptide (True7, Sigma-Aldrich, St Louis, MO, USA) were gelled with 10,000 cells according to manufacturer’s recommendations, in triplicate, in 96 well plates, for 14 days, with medium change every 3 days. Spheroid formation was documented using EvosXL phase-contrast microscope.

*Live-dead assay*–Cell viability was assessed by live/dead assay, using fluorescein diacetate (F1303, Invitrogen, Waltham, MA, USA) for living cells and propidium iodide (PI, P4170 Sigma-Aldrich, St Louis, MO, USA) for dead cells. A mixture of 80 µg/mL fluorescein diacetate and 200 ug/mL PI was freshly prepared, before each experiment, in serum-free medium. Then, 100 µL of the mixture was added to each well, and cells were incubated for 30 min in a cell incubator, with 5% CO_2,_ at 37 °C. The stain mixture was removed and cells washed 3 times in PBS (10 min each wash). Viability of cells was documented using an Evos FL microscope and the overlay of fluorescent channels was obtained using Evos software.

## Figures and Tables

**Figure 1 gels-09-00051-f001:**
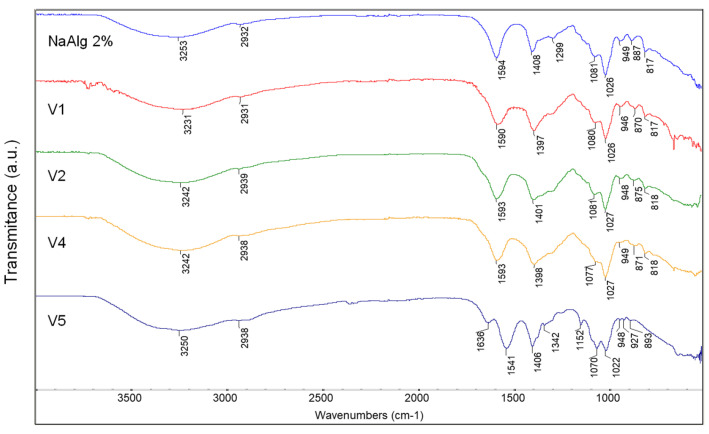
The FT-IR spectra of NaAlg 2%, V1, V2, V4, and V5 amorphous hydrogels.

**Figure 2 gels-09-00051-f002:**
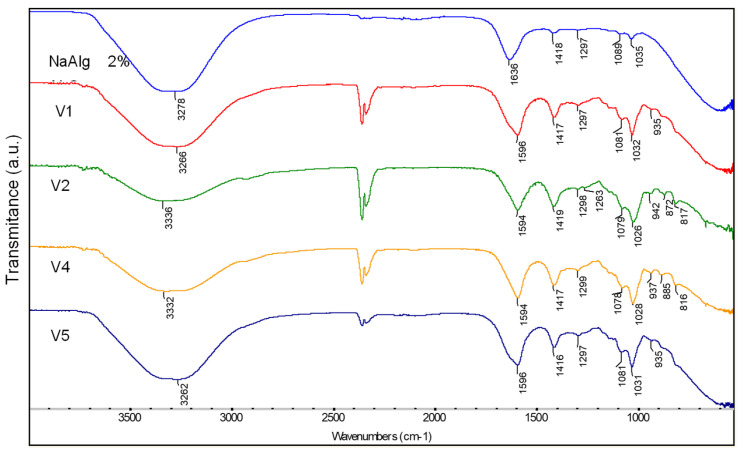
The FT-IR spectra of NaAlg 2%, V1, V2, V4, and V5 hydrogels crosslinked with calcium chloride.

**Figure 3 gels-09-00051-f003:**
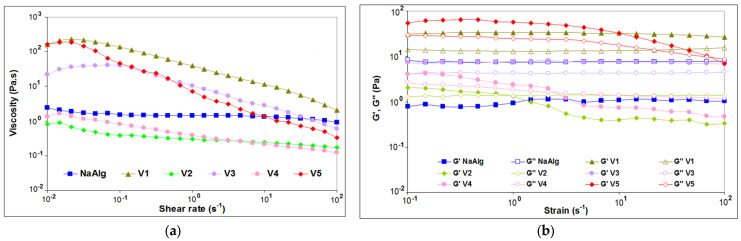
Rheological characterization: (**a**) viscosity as a function of shear rate for prepared alginate-based hydrogels at different pH and 37 °C and (**b**) dependence of G’ and G” moduli on the amplitude strain at 37 °C for the alginate-based hydrogels at different pH.

**Figure 4 gels-09-00051-f004:**
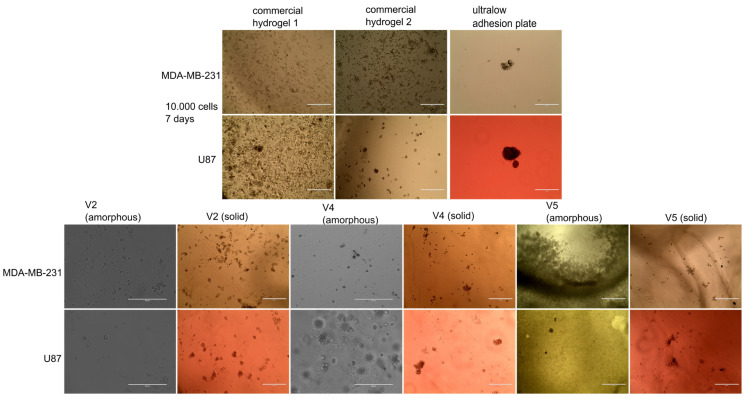
Microscope images for spheroids formed at 7 days in the developed gels. 10,000 cells of MDA-MB-231 and U87 were seeded in amorphous and solid hydrogels and incubated for 7 days in their respective complete media, with medium change on the 3rd and 6th day. 10×, phase contrast. Scale bar 400 µm.

**Figure 5 gels-09-00051-f005:**
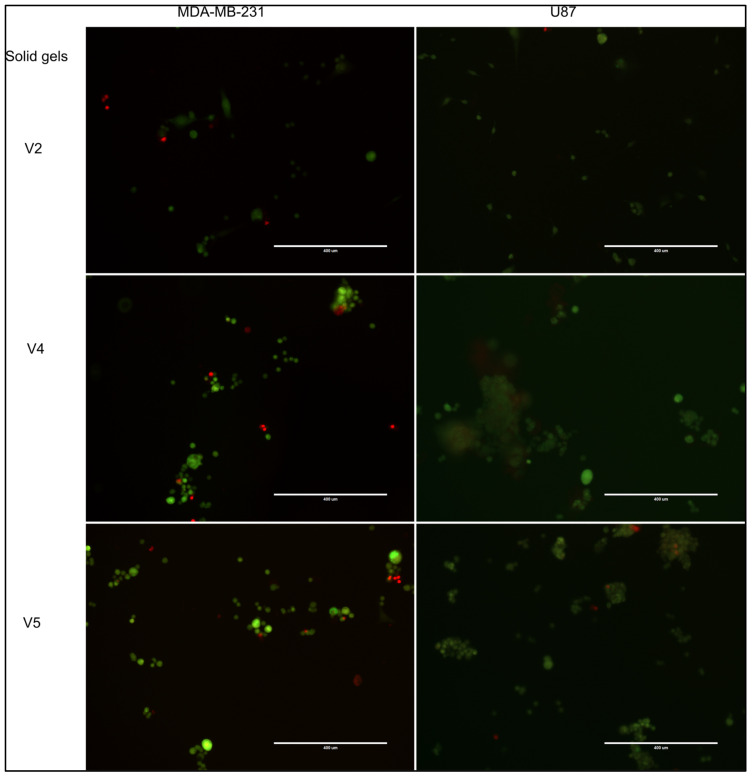
Viability of cells within spheroids was tested using live/dead fluorescent assay. Living cells are highlighted in green with fluorescein and dead cells in red with propidium iodide. 10× magnification. Scale bar 400 µm.

**Figure 6 gels-09-00051-f006:**
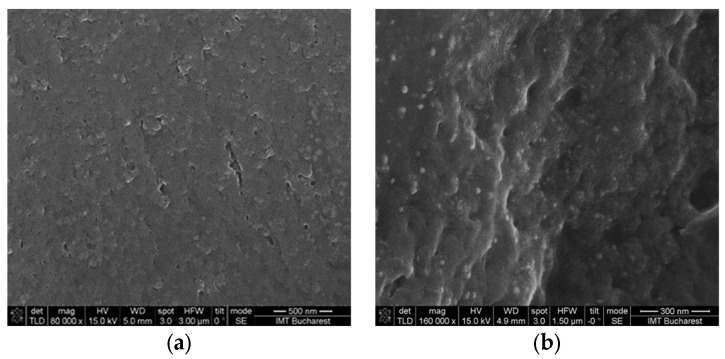
Morphology of the of hydrogels crosslinked with CaCl_2_, after 5 days of drying at room temperature: **(a)** SEM image of V2 hydrogel and **(b)** SEM image of V4 hydrogel.

**Figure 7 gels-09-00051-f007:**
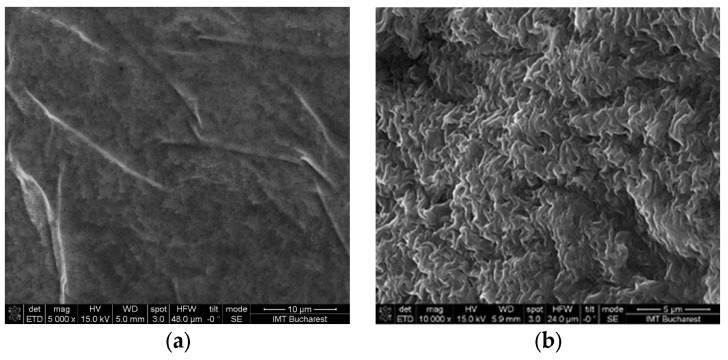
Morphology of the V5 hydrogel after 5 days of drying at room temperature: **(a)** SEM image of amorphous hydrogel and **(b)** SEM image of crosslinked hydrogel.

**Table 1 gels-09-00051-t001:** The composition of obtained hydrogels.

Experimental Variants (Code)	Sodium Alginate (2%) (%)	Aloe Vera Powder (1%) (%)	Chitosan (1%) (%)	Glucose (1%) (%)	Additional Treatment
V1	70.6	29.4	-		pH 5
V2	70.6	29.4	-		pH 12
V3	70.4	29.3	-	0.3	pH 5
V4	70.4	29.3	-	0.3	pH 12
V5	50	-	50	-	pH 5
V6	41.4	17.2	41.4	-	pH 5
V7	41.4	17.2	41.4	-	pH 12

## Data Availability

The data that support the findings of this study are available from the corresponding authors upon reasonable request.

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
