# Peer review of "Assessing Polysaccharides/Aloe Vera–Based Hydrogels for Tumor Spheroid Formation"

_gels, 2023, doi:10.3390/gels9010051_

Round 1
Reviewer 1 Report
The manuscript "Studying the tumor spheroids development capacity of polysaccharides / Aloe vera - based hydrogels" is overall good and covers important area. The authors focused on the development of inexpensive, easy to use and reliable scaffolds, based on polysaccharides and aloe vera, as a potential tumor spheroid formation system. They put in great effort and were successful in advancing the field of study. The methodology was picked carefully using many highly developed instruments. I suggest the following minor corrections:
Regarding Abstract: Please include significant findings with values.
line 97: I suggest to rewrite the sentence in this way; Alginate is a natural polysaccharide polymer.....
lines 142-143: Please clarify the sentence begins with "The plates with the hydrogels...
lines 568-574: Please clarify!
Figure 4: the authors need to improve the quality of this figure.
Author Response
Answer for Reviewer 1,
Thank you for evaluating our study, we also thank you for your comments and suggested guidance.
In what follows, we will respond to your observations.
- English has been revised (see the track-change in the attached document)
- The summary has been redone
In vitro tumor spheroids have proven to be useful 3D tumor culture models for drug testing, molecular mechanism of tumor progression and cellular interactions. There-fore, there is a continuous search for their industrial scalability and routine preparation. Considering that hydrogels are promising systems that can favor the formation of tumor spheroids, our study aimed to investigate and develop less expensive and easy to use amorphous and crosslinked hydrogels, based on natural compounds such as sodium alginate (NaAlg), Aloe vera gel (AV) powder and chitosan (CS) for tumor spheroids formation. The ability of the developed hydrogels as a potential spheroid forming system was evaluated using MDA-MB-231 and U87MG cancer cells. Spheroid abilities were influenced by the pH, viscosity and crosslinking of hydrogel. Addition of either AV or chitosan to sodium alginate increased the viscosity at pH 5, resulting amorphous hydrogels with a strong gel texture, as shown by rheologic analysis. Only the chitosan -based gel allowed formation of spheroids at pH 5. Among the variants of AV- based amorphous hydrogels tested, only hydrogels at pH 12 and with low viscosity promoted the formation of spheroids. The crosslinked NaAlg/AV, NaAlg/AV/Glucose and NaAlg/CS hydrogel variants favored more efficient spheroid formation. Additional studies would be needed to use AV in other physical forms and formulation hydrogels, as the current study is an initiation, in evaluating the potential use of AV gel in tumor spheroid formation systems.
- I modified the title
“Assessing polysaccharides / Aloe vera - based hydrogels for tumor spheroid formation”
- I noted the scale for the bars in the figures, in the legend of figure 4.
- I corrected the sentence that starts with “Alginate is a natural polysaccharide polymer……
“Alginate is a natural polysaccharide polymer, extracted from brown algae, previously used in drug delivery, wound dressing, tissue engineering and 3D cell culture setups. Starting from the aqueous solution of alginate, hydrogels can be obtained by ionic and covalent crosslinking [29]. “ (row 116-119)
- I clarified the sentence that starts with The plates with the hydrogels thus ….. (row 142-143), after reorganizing the body of the article with ‘Hydrogels were sterilized by 1h exposure to a UV-C lamp, then kept at 2-8°C degrees until further use”. (row 412-413)
- I have attached the references from 568-574.
Thank you!
Best regards,
Preda Petruta -National Institute for Research and Development in Microtechnologies—IMT Bucharest, 126A Erou Iancu Nicolae, 077190 Bucharest, Romania

Reviewer 2 Report
Please, See the comments in the attached manuscript file.

Author Response
Answer for Reviewer 2,
Thank you for evaluating our study, we also thank you for your comments and suggested guidance.
In what follows, we will respond to your observations.
- English has been revised (see the track-change in the attached document)
- The summary has been redone
In vitro tumor spheroids have proven to be useful 3D tumor culture models for drug testing, molecular mechanism of tumor progression and cellular interactions. Therefore, there is a continuous search for their industrial scalability and routine preparation. Considering that hydrogels are promising systems that can favor the formation of tumor spheroids, our study aimed to investigate and develop less expensive and easy to use amorphous and crosslinked hydrogels, based on natural compounds such as sodium alginate (NaAlg), Aloe vera gel (AV) powder and chitosan (CS) for tumor spheroids formation. The ability of the developed hydrogels as a potential spheroid forming system was evaluated using MDA-MB-231 and U87MG cancer cells. Spheroid abilities were influenced by the pH, viscosity and crosslinking of hydrogel. Addition of either AV or chitosan to sodium alginate increased the viscosity at pH 5, resulting amorphous hydrogels with a strong gel texture, as shown by rheologic analysis. Only the chitosan -based gel allowed formation of spheroids at pH 5. Among the variants of AV- based amorphous hydrogels tested, only hydrogels at pH 12 and with low viscosity promoted the formation of spheroids. The crosslinked NaAlg/AV, NaAlg/AV/Glucose and NaAlg/CS hydrogel variants favored more efficient spheroid formation. Additional studies would be needed to use AV in other physical forms and formulation hydrogels, as the current study is an initiation, in evaluating the potential use of AV gel in tumor spheroid formation systems.
- The introduction section is too long, and contains redundancies.
Answer: I reduced the introduction, and spheroid application is found in the phrase
“Due to mimicking native tumor tissue, the multicellular tumor spheroids (MCTS) are the current approach used in vitro as 3D cancer cells culture models for therapeutic strategies (e.g, drug therapy testing, to study tumor progression under different condi-tions such as early stage and avascular tumor, and also to evaluate the release of solu-ble mediators that provide clues for immunosurveillance) [5] and genetic screening and editing tumor cells for personalized medicine [6, 7].”
- Revise citation format….. Hence, the ability of oxygenating microgels formed utilizing perfluorocarbon (PFC) modified chitosan was studied by Patil et al. 2020 which facilitated the growth of larger human cell-based spheroids whit higher oxygen pressures into spheroids [19].
Answer: Hence, the ability of oxygenating microgels formed using perfluorocarbon (PFC) modified chitosan was studied by Patil et al., which facilitated the growth of larger human cell-based spheroids with higher oxygen pressures into spheroids [21].
- based on was modified in Hydrogels based of alginate/gelatin/Matrigel…. with Hydrogels based on alginate/gelatin/Matrigel were…..
Answer: thank you!
- Paragraphs from line 88-120 should be presented in the discussion section.
Answer: were moved
- Details without any data
Answer:
We add a supplementary material regarding behavior of cells in one of the variants, where, although cells were viable, no spheroids were formed (Supplementary S3).
- 1 – image scale bar is not clear..
Fig 4 is now at 600 dpi resolution. We added the following information in the Legend: scale bar 400µm.
- Fig 4 - image scale bar is not clear
Fig 5 is now at 600 dpi resolution. The image should be clear and can be zoomed in without pixelation. We added the following information in the Legend: scale bar 400µm.
- A discussion based on the previous studies should be presented
Answer:
Previous studies reported use of alginate [57, 58], or alginate-chitosan scaffolds [59] to generate viable glioblastoma spheroids. Although the number of seeding cells was much lower in the present study (104, as opposed to 5x104-106), the spheroid size was larger. The use of AV could provide additional benefits for cells, as it has been re-ported to have bioactive components which favors cell proliferation, migration and angiogenesis [60-61]. In addition, it has recently been demonstrated that glucomannan from AV gel facilitated the regeneration of intestinal epithelial cells [62]. Alginate has been proven as a viable solution also for MDA-MB-231 spheroids [63], but chitosan-alginate scaffolds are only beginning to be investigated [64]. So far, no data regarding alginate-Aloe vera scaffolds has been reported regarding MDA-MB-231 and U87MG spheroid formation.
- Authors should correlate the results of FTIR, SEM and rheology with the properties of the prepared hydrogel beads in term of cell viability and tissue attachment –
Answer: at conclusions
Moreover, it was observed by rheological analysis that the hydrogel texture is pH dependent. The fact that the variants containing AV at pH 12 (liquid like) favored the spheroids formation (for U87 and MDA-MB-231 cells) compared to their counterparts at pH 5, could be associated with the strong gel texture of the latter ones, probably could not allow oxygen access and cell migration making the development of cell cul-ture and implicitly spheroids impossible. Although it presented a gel texture, crosslink-ing with CaCl2 allowed obtaining a porous hydrogel for mixture with pH 5 that con-tained alginate and chitosan, so the porosity facilitating in this case the good formation of spheroids compared to the amorphous version.
- The conclusions have been modified
- Material and method – Revise
Answer:
Revised to: “ Cell lines and spheroid culture - Human breast carcinoma cells MDA-MB-231 (ATCC: HTB-26) and human glioblastoma cells U87MG (ATCC HTB-14) were routinely cultured in an incubator, at 5%CO2 and 37°C. Every other day the cell culture medium was changed with RPMI-1640 supplemented with 10% fetal bovine serum or DMEM/F12 supplemented with 10% fetal bovine serum, respectively. For spheroid formation, cells were trypsinized, centrifuged, counted and 10 000 cells were incubated in each well, containing various gel compositions.
Do not use FDA for abbreviating this reagent!
Abbreviation deleted
Live-dead assay -Revise
The sentence was revised as follows:
A mixture of 80µg/mL fluorescein diacetate and 200ug/mL PI was freshly prepared, before each experiment, in serum-free medium. 100µl of the mixture was added to each well, then cells were incubated for 30 minutes in a cell incubator, with 5%CO2, at 37°C. The stain mixture was removed and cells washed 3 times in PBS (10 minutes each wash). Viability of cells was documented using an Evos FL microscope and the overlay of fluorescent channels was obtained using the Evos software.
Hydrogel Formulation
The gel composites were developed by stirring mixtures based on sodium alginate solution (2%), chitosan solution (1%), AV gel powder solution (1%) and glucose solution (1%) respectively. The mixtures were obtained at room temperature (25 oC), the acid pH (pH = 4.5-5) of the biocomposites (gels) was corrected with 3M NaOH solution. The obtained formulations were sterilized by exposure to UV lamp for 30 min and stored in a refrigerator (2–8 oC) until further use. The composition of developed hydrogels is presented in Table 1. It should be noted that all hydrogel variants were obtained both as amorphous and solid gels by crosslinking with CaCl2 solution (1%).
The preparation of the 2% NaAlg solution was carried out in water by stirring 2h at 300 rpm and 80 oC. Similar conditions were used for obtaining 1% CS solution in 1% acid acetic. AV solution (1%) was obtained by solubilization of commercial AV powder in water, followed by gentle stirring for 10 minutes at room temperature.
From more than 20 variants of formulations, with different composition and pH, 7 alginate-based hydrogels were selected, combined in different proportion with AV and/or chitosan, and/or glucose.
Table 1. The composition of obtained hydrogels.
Experimental Variants (Code) |
Sodium Alginate (2%) (%) |
Aloe vera powder (1%) (%) |
Chitosan (1%)
(%) |
Glucose (1%) (%) |
Additional treatment |
V1 |
70.6 |
29.4 |
- |
|
pH 5 |
V2 |
70.6 |
29.4 |
- |
|
pH 12 |
V3 |
70.4 |
29.3 |
- |
0.3 |
pH 5 |
V4 |
70.4 |
29.3 |
- |
0.3 |
pH 12 |
V5 |
50 |
- |
50 |
- |
pH 4.5 |
V6 |
41.4 |
17.2 |
41.4 |
- |
pH 5 |
V7 |
41.4 |
17.2 |
41.4 |
- |
pH 12 |
Both amorphous and crosslinked gels (with 1% CaCl2 solution) were obtained. All forms of hydrogels were transparent, with a slight tendency to yellow in the case of gels at pH 12. 2 ml of gel was distributed in 24-well sterile plates followed by addition of CaCl2 solution. After 30 minutes of CaCl2 solution action the crosslinked hydrogels were washed three times with deionized water. Hydrogels were sterilized by 1h exposure to a UV-C lamp, then kept at 2-8°C degrees until further use.
- How did you prepare hydrogel samples for FTIR studies?
Answer:
4.2. Structural, Morphological and Reological Investigation of Hydrogels
The preparation of the amorphous and solid hydrogels for the morphological and structural analysis consisted in drying the samples at room temperature for 5 days. In the case of the amorphous hydrogel variants, a film was obtained by depositing the gel on the glass slide and then drying. The crosslinked hydrogels samples were also ob-tained by drying in the wells. The morphological analysis of the obtained hydrogels was performed by using the field emission scanning electron microscope (FE-SEM) Nova NanoSEM 630 (FEI Company, USA). The Fourier-transform infrared spectra (FTIR) of obtained hydrogels were collected by using the ATR device of the Nicolet i-S10 FTIR spectrometer (Thermo Scientific USA). The device was equipped with a dia-mond crystal and the spectra were recorded in the 4000-525 cm-1 spectral region at 4 cm-1 resolution with 32 scans.
Thank you!
Best regards,
Preda Petruta -National Institute for Research and Development in Microtechnologies—IMT Bucharest, 126A Erou Iancu Nicolae, 077190 Bucharest, Romania

Round 2
Reviewer 2 Report
1. Correct CO2 to CO2 in line 448, and in the whole manuscript.